# A novel statistical method predicts mutability of the genomic segments of the SARS-CoV-2 virus

Amir Hossein Darooneh ⬤, Michelle Przedborski ⬤ and Mohammad Kohandel* ⬤

Department of Applied Mathematics, University of Waterloo, Waterloo, ON, Canada

SARS-CoV2; Mutability; Statistical Analysis; Word Ranking

**Author for correspondence:**
*Mohammad Kohandel,
E-mail: kohandel@uwaterloo.ca

## Abstract

The SARS-CoV-2 virus has made the largest pandemic of the 21st century, with hundreds of millions of cases and tens of millions of fatalities. Scientists all around the world are racing to develop vaccines and new pharmaceuticals to overcome the pandemic and offer effective treatments for COVID-19 disease. Consequently, there is an essential need to better understand how the pathogenesis of SARS-CoV-2 is affected by viral mutations and to determine the conserved segments in the viral genome that can serve as stable targets for novel therapeutics. Here, we introduce a text-mining method to estimate the mutability of genomic segments directly from a reference (ancestral) whole genome sequence. The method relies on calculating the importance of genomic segments based on their spatial distribution and frequency over the whole genome. To validate our approach, we perform a large-scale analysis of the viral mutations in nearly 80,000 publicly available SARS-CoV-2 predecessor whole genome sequences and show that these results are highly correlated with the segments predicted by the statistical method used for keyword detection. Importantly, these correlations are found to hold at the codon and gene levels, as well as for gene coding regions. Using the text-mining method, we further identify codon sequences that are potential candidates for siRNA-based antiviral drugs. Significantly, one of the candidates identified in this work corresponds to the first seven codons of an epitope of the spike glycoprotein, which is the only SARS-CoV-2 immunogenic peptide without a match to a human protein.

## Introduction

Currently, the world is in an unprecedented state of crisis due to the COVID-19 pandemic, which has affected the health and social behaviour of humans, as well as taken a considerable toll on the global economy. The cause of this deadly disease is the SARS-CoV-2 virus, which is a novel member of the Coronavirus family whose origin and chain of infection to humans still remains uncertain (Andersen *et al.,* 2020; Burki, 2020). Researchers around the globe are racing to develop therapeutic strategies to help mitigate the pandemic, including novel antiviral drugs, appropriate combinations of existing pharmaceuticals, and SARS-CoV-2 vaccine candidates.

To ensure sustained efficacy, it is imperative that novel therapeutics target the conserved parts of the viral genome (Hermann, 2016; Hu *et al.,* 2020; Srinivasan *et al.,* 2020) since these segments are not considerably affected by viral mutations. Several methods that are based on sequence alignment currently exist to find the conserved parts of a viral genome (Stojanovic, 1999; Nagar and Hahsler, 2013; Wiltgen, 2019). These methods involve performing pairwise alignment between the predecessor and a descendant sequence. While this enables the characterisation of the viral mutations, these methods require an extensive number of sequences that are rooted from the same ancestor and must be collected over time. Thus, when facing a novel virus or pathogen that has the potential to lead to a widespread epidemic or a global pandemic, this waiting process impedes the rapid development of targeted therapeutics that could have a critical impact on the case fatality rate or the magnitude of the outbreak. Importantly, the mutation information can seldom be extracted directly from the mutational changes observed in other members in the virus family. For example, the SARS-CoV-2 virus has undergone considerably different changes in comparison with other members of the Coronaviridae family, such as its cousins SARS-CoV and MERS (Naqvi *et al.,* 2020; Wu *et al.,* 2020).

Due to the urgency in controlling a highly infectious pathogen, it is imperative to have a method for extracting mutational information as quickly as possible so that therapeutic targets can be promptly identified. A method that can extract the mutational propensity of different segments of the whole genome directly from an ancestral sequence is thus ideal for the rapid development of targeted therapeutics when a novel pathogen is identified. Here we introduce one such method that is similar to the keyword detection techniques in text-mining. Using this simple approach, an arbitrary genomic segment is assigned an importance value based on its repetition and spatial distribution within the ancestral genome. The importance values can then be used to estimate the mutability of segments in the whole genome, where the conserved or low mutation parts are those with a high importance value. To demonstrate the validity of this approach, we

apply the method to the SARS-CoV-2 reference genome (NCBI, 2020) and show that the segments that are identified as important strongly correlate with the conserved sequences that are identified through standard mutational analysis of nearly 80,000 complete genome sequences for the virus. Importantly, we further use the approach to identify conserved segments of six and seven codons in the SARS-CoV-2 genome that are potential candidates for stable siRNA-based targeted drugs.

The manuscript is organised as follows. In section 'Methods', we describe the empirical SARS-CoV-2 data that are used in the study and introduce the text-mining method developed in the work. In section 'Results', we present the results of standard mutation analysis of the SARS-CoV-2 viral genomes and compare these findings with the results from the text-mining method applied to the reference genome. Finally, in section 'Discussion', we discuss the implications of our results and suggest potential future research directions.

## Methods

### Curation and preparation of genomic data

In this study, nearly 80,000 complete genome sequences for SARS-CoV-2 were curated from two public repositories and then subsequently analysed. First, 7,031 complete genome sequences were obtained from NCBI (2020). Thereafter, we obtained 74,750 genome sequences from GISAID (2020) which most of them (99.8%) contained more than 29,000 nucleotides and were considered to be complete genomes in this work.

The reference sequence, NC_045512.2, was also obtained from NCBI (2020). This is the first genomic sequence for SARS-CoV-2, whose origin is Wuhan, China, and it was made publicly available on NCBI in January 2020. The reference sequence is comprised of 29,903 nucleotides. According to the NCBI database, the sequence contains 28 coding regions that specifically code the virus proteins. These coding regions are encompassed in 10 genes: ORF1ab, S, ORF3a, E, M, ORF6, ORF7ab, ORF8, N and ORF10. These 10 genes occupy approximately 97.86% of the virus genome. The longest gene is ORF1ab, with a length of 21,290 nucleotides, while the shortest gene, ORF10, has length 117 nucleotides.

The genomic sequences from both data sets were aligned with the reference sequence, NC_045512.2, using the NCBI's BLASTN 2.6.0+ software (Zhang *et al.*, 2000). To avoid overlap between the two data sets, and to facilitate comparison of the results, the subsequent genomic analysis was performed independently for each data set. Albeit in most instances we proceed with reporting the results for the larger data set (GISAID).

Based on alignments, any certain change in the genome sequences with respect to the reference sequence, including nucleotide insertions, deletions and substitutions, were extracted into a master file. This master file contained the mutation information for each sequence, as well as the specific nucleotide change and its position in the reference genome, and is the foundation for the empirical analysis.

During the empirical analysis, we investigated the number and type of mutations in the SARS-CoV-2 genome. We looked for changes at the individual nucleotide level, including the distribution of changes at different positions in the genome. We also investigated changes at the codon level, including insertion, substitution and deletion mutations, and whether these mutations cause changes at the protein level.

### Word ranking technique

The arrangement of nucleotides in a genome sequence is not purely random. This is because physical and chemical interactions between nucleotides determine which nucleotide is more likely to be a neighbour of another nucleotide (Alberts *et al.*, 2002; Chen *et al.*, 2016). These interactions thus give rise to order in the sequence; however, randomness is introduced by thermal fluctuations, environmental interactions and non-equilibrium conditions arising during replication processes. The competition between order and disorder leads to some segments of the sequence being more stable, or important, than others. This observation is very similar to patterns in written texts, in which some words are more important and responsible for conveying the meaning of a passage of text, while others are common words. Consequently, it seems only natural to develop text-mining-based techniques to identify important parts in a specific genome sequence.

We consider a text as a one-dimensional array. The appearances of a specific word are occupied distinct positions in this discrete space. In random text, words are distributed uniformly because there is no preference for placing a word in proximity to another word and the position of a word is independent of the position of other words. Therefore, in every part of text there is a non-zero probability for finding a certain word. In contrast, in natural text, the positions of words are determined based on the grammatical rules and the context of the text, thus the position of each word strongly depends on the position of other words. For genomic sequences, the overall functionality of the sequence is synonymous with the meaning of a passage of text, and the chemical and physical interactions between nucleotides or codons are analogous to grammatical rules. The existence of short-range and long-range order causes the distribution pattern of a word to deviate from uniformity and to be clustered. The important words are more clustered than common words. By randomly shuffling the words in a passage of text, the meaning is lost and grammatical rules are also violated. Importantly, shuffling does not considerably alter the pattern of words that are distributed near uniformly, whereas the distribution of clustered words experiences a drastic change. This observation allows us to consider the clustering as a measure of importance in addition to the word frequency.

Several methods exist for ranking the distinct words of a text according to their importance (Najafi and Darooneh, 2015). All these methods identify the clustering of a word in the text. Here we develop a new approach for characterising genome segments that is based on the frequency of occurrence of the segment in the whole genome, as well as its closeness to boundaries and clustering within the genome, to associate an importance value to different genomic 'words' for SARS-CoV-2.

The starting point in this approach is defining what is considered a 'word' in the genomic sequence. Here we consider codons, which are three-nucleotide sequences that encode for amino acids, as the words. The position of the codon is taken to be the location of its first nucleotide in the sequence. Like the keywords in text, we assume that the significant codons form clusters. Furthermore, clustering near region boundaries is assumed to be more important than other places in the genomic sequence (Parmley and Hurst, 2007; Esposito *et al.*, 2010; Chaney *et al.*, 2017). To quantify these properties, we define the eccentricity $e(w)$ for a word (codon) $w$, as follows:

$$e(w) = \frac{1}{R} \sum_{r=1}^{R} \sum_{i=1}^{f_r(w)} (x_i(w) - m_r)^2, \qquad (1)$$

where $R$ is the number of regions in the sequence and $f_r(w)$ is the frequency of occurrence of $w$ in the $r$th region. Additionally, $x_i(w)$ is the $i$th position of the word in the region and $m_r$ is the position of the first quarter of the $r$th region. This convention implies that clustering at the end of a region is more important than at the beginning. When a coding region is formed by joining several disconnected smaller regions, the eccentricity from the first quarter of all smaller regions must be taken into account. However, this definition of the eccentricity will not be able to capture clustering near $m_r$ as required.

In addition to the eccentricity, the importance of a codon also depends on its frequency of occurrence in the genome. Indeed, many researchers believe that bias in repetition of codons has biological consequences (Angov, 2011; Lauring *et al.,* 2012; Zhou *et al.,* 2016). Thus, the frequency of occurrence should be combined with eccentricity to have a unified rule for assigning importance to codons. We suggest the following formula for calculation of the importance $i(w)$ of codon $w$:

$$i(w) = \ln\left(1 + \sqrt{\frac{f(w)}{\sum_w f(v)}}\right) \times \ln\left(1 + \frac{e(w)}{\sum_v e(v)}\right), \qquad (2)$$

where $f(w)$ is the frequency of occurrence of codon $w$ and $e(w)$ is its eccentricity, calculated using Eq. (1). This form was chosen for the importance because, by normalising both the frequency and the eccentricity, their values become comparable to each other. The square root appears in the first logarithm to account for the fact that the frequency values are distributed over a larger scale than the eccentricity values. Based on the empirical data, taking the square root of normalised frequency reduces the variation in the frequency of different codons to one order of magnitude, like their eccentricity variation. Thus, incorporation of the square root inhibits the codon frequency from dominating the importance. The logarithm is further used to reduce large differences. Moreover, by multiplication of the two logarithmic expressions, all combinations between frequency and eccentricity are taken to account. It should be noted that the importance can be scaled by an arbitrary factor. Thus, to draw conclusions about which codons are most important within a given genome, it is necessary to examine how a given codon compares with all other codons in that genome.

In this work, we compute the eccentricity of codons over 28 coding regions in the SARS-CoV-2 genomic sequence. Some coding regions have overlap, therefore, a codon-position may contribute to the calculation of the importance more than one time through the eccentricity.

## Results

### Empirical analysis characterises SARS-CoV-2 viral mutations

We identified 57,939 mutations in the curated NCBI data set and 674,800 in the curated GISAID data set. In the NCBI data set, 48,554 or 83.8% of the mutations occurred in coding regions, while 561,195 or 83.2% were in coding regions for the GISAID data set. The mutation rate per nucleotide per generation, calculated as the number of mutations per number of sequences and per sequence length, is approximately $2.75 \times 10^{-5}$ and $3.02 \times 10^{-5}$, respectively, for the NCBI and GISAID data sets. Thus the probability of mutation in the SARS-CoV-2 genome is low in comparison with other RNA viruses (Drake *et al.,* 1998), which accounts for the stability of the SARS-CoV-2 virus and the emergence of the recent pandemic (Duffy, 2018; Peck and Lauring, 2018).

The frequency of nucleotide occurrence in a genome is a basic specification for any genomic sequence, and it is related to the effective energy usage by organisms in their duplication process (Chen *et al.,* 2016). In Fig. 1, we present the probability of occurrence of each nucleotide in the reference sequence NC_045512.2, calculated as the fraction of the whole genome comprised of each nucleotide. As we see from Fig. 1, each nucleotide type does not appear with the same frequency of occurrence. In particular, Thymine (T) occurs most frequently 32.08% in the reference sequence, while Cytosine (C) appears least often, 18.36%. In addition, in Fig. 1, we plot the fraction of mutated nucleotides, which evidently varies significantly among the different bases. These results indicate that mutations do not occur randomly, since in the latter case, each nucleotide type would have the same probability of mutation and occurrence. Interestingly, Cytosine has a significantly higher probability of mutation, despite being the lowest occurring nucleotide (54.51% in the NCBI and 49.23% in the GISAID sequences). In contrast, Thymine, which occurs most frequently, has the lowest probability of mutation, 8.17 and 8.41% for the NCBI and GISAID datasets, respectively. The details are reported in Supplementary Information 1.

The percentage of different mutations in the SARS-CoV-2 genome that are observed for each nucleotide type is shown in Fig. 2. Overall the pattern of evolution, that is GC →AT, is apparent in the virus mutations (Greenbaum *et al.,* 2008). Interestingly, we see that the results for both the NCBI and GISAID data sets are quite similar, with slight differences in the distributions for Cytosine, Thymine, and nucleotide insertions. Importantly, we see from this figure that for each nucleotide, there is a distinct destination for substitution mutations. Specifically, the most probable mutations

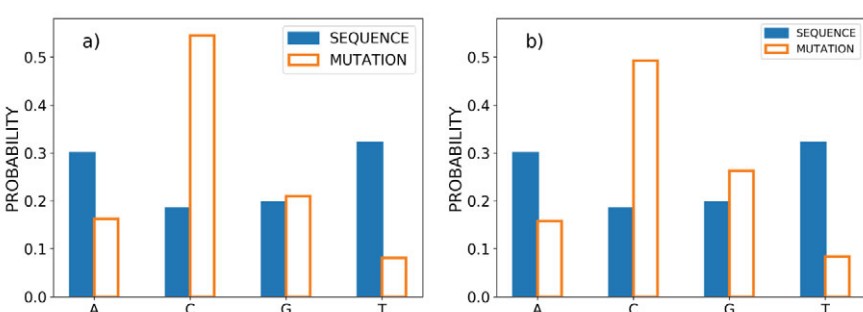

**Fig. 1.** The probability of appearance of each nucleotide (A, Adenine; C, Cytosine; G, Guanine; T, Thymine) in the reference SARS-CoV-2 genome sequence (blue solid bar), and the fraction of mutated nucleotides (orange bar), for the (*a*) NCBI data set and (*b*) GISAID data set.

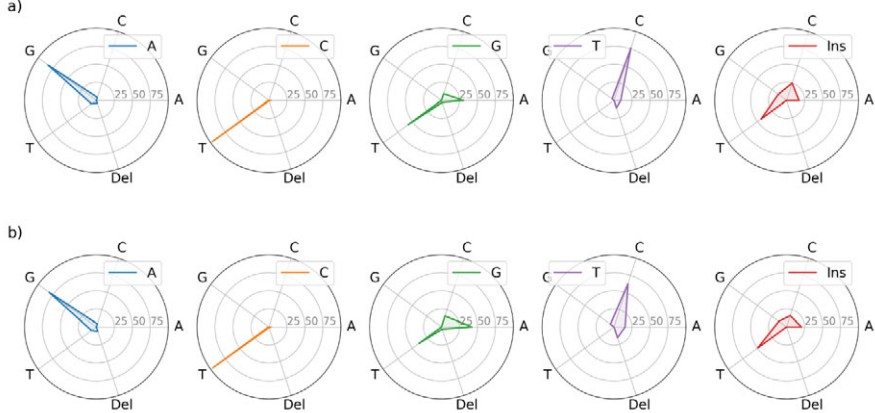

**Figure 2.** Percentage of different mutations observed for each nucleotide (A, C, G, T) for (*a*) the NCBI data set and (*b*) the GISAID data set. Axis labels reference nucleotide substitutions, 'Del' refers to deletion events, and 'Ins' are nucleotide insertions.

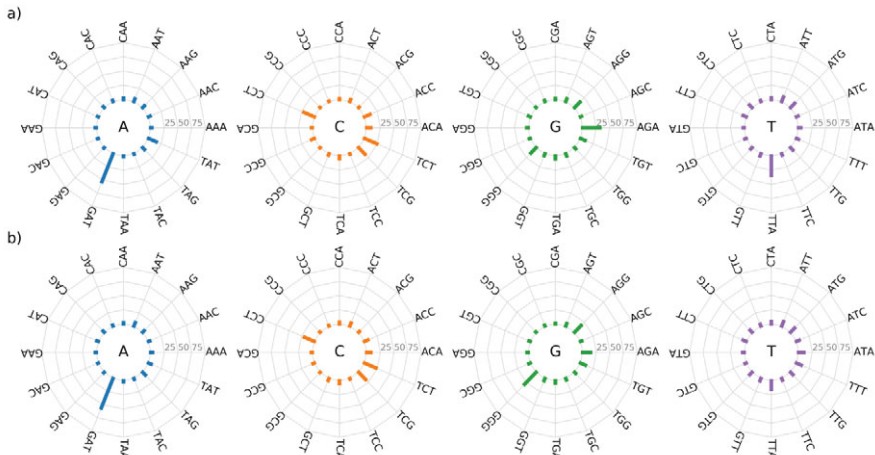

**Figure 3.** The percentage of mutations that occur among different possible three-nucleotide sequences. For each sequence, the mutation occurs in the central nucleotide, as indicated at the centre of each plot. Results are shown for (*a*) the NCBI data set and (*b*) the GISAID data set.

are: A →G (83.16 and 81.18%), C →T (96.72 and 95.95%), G →T (57.01 and 38.77%) and T →C (76.47 and 62.70%). In contrast, we see that for the insertion mutations, all nucleotide types have a considerable probability, while Thymine has the highest chance (43.75 and 49.76%). The first and second values in the parentheses are extracted from the NCBI and GISAID datasets, respectively. The details are reported in Supplementary Information 1.

All mutated nucleotides have a neighbour on both their left and right sides. To determine whether all the corresponding nucleotide sequences have equal propensity for mutation, we plot the percentage of mutations that are observed for each of the 16 possible nucleotide sequences, for each mutated nucleotide, in Fig. 3. We see that for all four mutated central nucleotides, the distribution is not uniform among the different nucleotide sequences. Importantly, this indicates that certain nucleotide sequences have a significantly higher propensity for mutation than others. For example, Cytosine is more likely to mutate when it is preceded by another Cytosine or a Thymine, with the highest probability occurring when it is also succeeded by a Thymine. Specifically, for TCT, the mutation percentage is 24.22 and 22.51% for the NCBI and

GISAID, respectively. Interestingly, we see from Fig. 3 that, overall, a nucleotide has a considerable probability of mutation if it has Guanine, Adenine or Thymine, but not Cytosine, in its immediate vicinity. Furthermore, we point out that there are slight differences in the results for the NCBI and GISAID data sets. The details are reported in Supplementary Information 1.

The nucleotide sequences in Fig. 3 each occur at several positions within the SARS-CoV-2 whole genome sequence. Though almost all such positions for each nucleotide sequence are observed to be mutated, the number of mutations is not uniform across all positions along the sequence. In Table 1, we report the top 10 positions with the highest frequency of substitution mutation observed in the GISAID data set. We see from Table 1 that some positions along the whole genome are indeed significantly more predisposed for mutation, with the top five positions comprising over 37% of all the mutations in the genome.

As mentioned above, most of the SARS-CoV-2 genome is occupied by codons, which are sequences of three nucleotides that code for specific amino acids during protein synthesis. Importantly, the codons and the corresponding protein structure of the SARS-

**Table 1.** Top 10 nucleotide positions with the highest probability of substitution mutation in the SARS-CoV-2 genomic sequence, based on the GISAID data set

| Position | Nucleotide | Repetition | Left neighbour | Right neighbour |
|----------|-----------|-----------|----------------|-----------------|
| 23403 | A | 57221 | G | T |
| 14408 | C | 56993 | C | T |
| 3037 | C | 56957 | T | T |
| 241 | C | 55901 | T | G |
| 28881 | G | 24173 | A | G |
| 28882 | G | 24125 | G | G |
| 28883 | G | 24117 | G | G |
| 25563 | G | 16264 | A | A |
| 1059 | C | 12614 | A | C |
| 11083 | G | 8286 | T | T |

CoV-2 virus may change due to mutations in the nucleotide sequences. To analyse the impact of the observed nucleotide mutations on the synthesis of viral proteins, we first examined the underlying distribution of codons in the SARS-CoV-2 reference sequence, NC_045512.2. Importantly, we found that there is a non-uniform distribution in the frequency of occurrence of each codon (see Supplementary Information 2, Fig. S1). We next determined the number of codon mutations in each nucleotide position of the whole genome, which is presented in Fig. 4 for the GISAID data set. We see from the figure that the distribution of mutations is non-uniform over the genome, displaying some background periodicities. Interestingly, several of the codon positions which include nucleotides with more than 50,000 mutations are identified in this figure. These nucleotides are in accordance with the results in Table 1.

Building on these results, we investigated the distribution in the number of codon mutations along the SARS-CoV-2 genome in the GISAID data set. We found that the distribution function behaves non-monotonically as a function of the number of mutations, and it peaks between approximately 8 and 32 codon mutations (see Supplementary Information 2, Fig. S2). Interestingly, the number of positions in the SARS-CoV-2 genome with a given number of

codon mutations increases linearly when the number of mutations is small ( $\lesssim 2^3$ ); however, when the number of mutations exceeds $\approx 2^3$ , the number of positions with a given number of codon mutations is inversely related to the mutation repetition number. This power law behaviour emphasises that the evolution of the SARS-CoV-2 genome is not a purely random process, but rather, it obeys some universal physical rules.

While we found that the evolution of the SARS-CoV-2 genome is not a purely random process, examining the total number of codon mutations does not give insight into the nature of the mutations that occur. To gain further insight into the specific codon mutations, we begin by examining the probability for different possible nucleotide-substitution codon changes, calculated based on the frequency of the observed mutation in the GISAID data set. The results are depicted in Fig. 5 and the details are also reported in Supplementary Information 1. Importantly, we see that the probability of each codon change is non-zero for only a small fraction of the possible codon changes. Interestingly, the codon changes with higher probability of mutation follows, on average, a series of single straight lines that originate at the top left and end at the bottom right of the plot. Each line corresponds to a specific position (first, second or third) in the codon that is mutated, and the mutation is repetitive. For example, for the lowest line emerging from the left of Fig. 5, at the codon GAA, the first nucleotide is mutated to an Adenine for the codons beginning with Guanine, and to a Cytosine for the codons beginning with a Thymine. The 10 most frequently observed nucleotide-substitution codon changes exhibited in Fig. 5 are listed in Table 2. Comparison of Tables 1 and 2 reveals that some of the nucleotide positions in the SARS-CoV-2 genome with the highest probability for mutation are not involved in codon changes, for example position 241.

As exhibited in Fig. 2, deletion mutations also occur in the SARS-CoV-2 genome. Deletion mutations, which result in the removal of segments of the genome, can cause some adjacent codons to merge into each other, potentially forming a new codon. In Table 3, we report the 10 most frequent deletion mutations that were observed in the SARS-CoV-2 genome in the GISAID data set. Importantly, we see that several of the most frequent deletion events lead to a new codon, which can change the encoded amino acid, and potentially lead to protein-level changes in the virus. We also found that there are very rare cases in which the number of adjacent nucleotide deletions is not a multiplicative factor of three. While rare, such deletions can dramatically change the protein sequence. In agreement with previous work (Mercatelli and Giorgi, 2020),

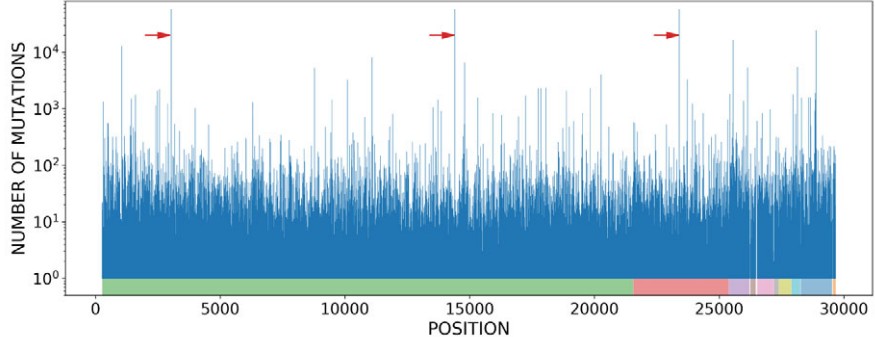

**Figure 4.** Number of codon mutations associated with each nucleotide position in the SARS-CoV-2 whole genome, according to the GISAID data set. The coloured rectangles in the bottom of the figure depict different gene regions. Three codon positions that include nucleotides with more than 50,000 mutations are identified by red arrows.

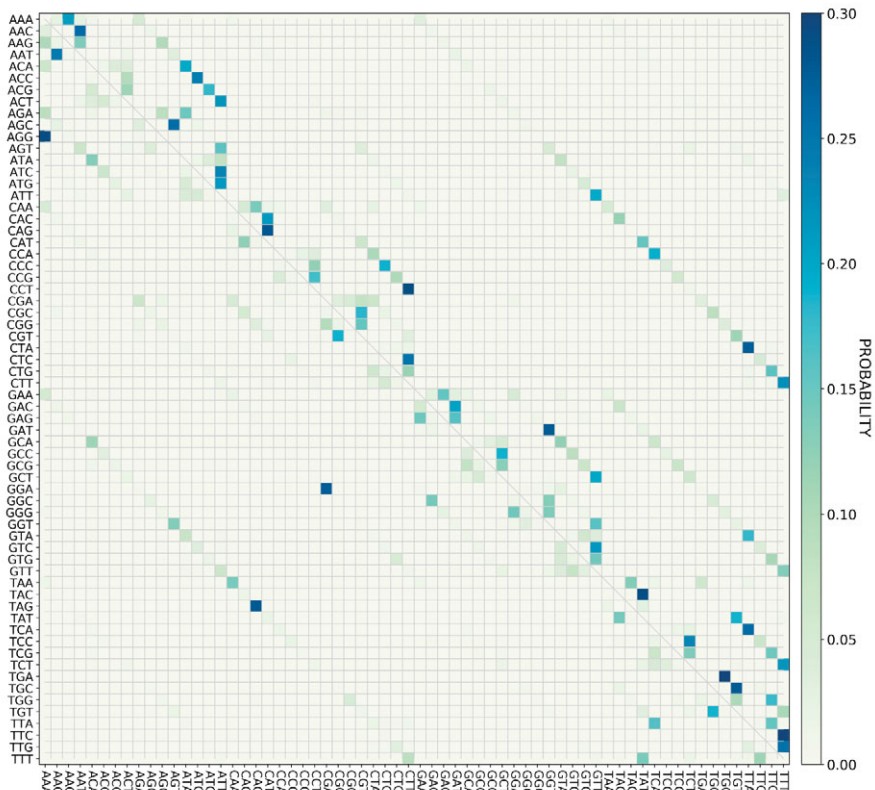

**Figure 5.** Probability of nucleotide-substitution codon changes in the SARS-CoV-2 genome, based on the GISAID data set. The *y*-axis corresponds to the origin codons in the reference genome and the *x*-axis is the destination codon. The codons are arranged in alphabetical order along each axis.

**Table 2.** The 10 most probable codon changes in the SARS-CoV-2 genome, according to the GISAID data set

| Position | Origin | Destination | Repetition |
|---|---|---|---|
| 23402 | GAT | GGT | 57221 |
| 14407 | CCT | CTT | 56966 |
| 3035 | TTC | TTT | 56956 |
| 28880 | AGG | AAA | 24088 |
| 28883 | GGA | CGA | 24085 |
| 25561 | CAG | CAT | 16223 |
| 1058 | ACC | ATC | 12605 |
| 11081 | TTG | TTT | 8173 |
| 14803 | TAC | TAT | 6524 |
| 28143 | TTA | TCA | 5385 |

**Table 3.** Top 10 most frequent deletion mutations in the SARS-CoV-2 genome, causing the merging or removal of codons, based on the GISAID data set

| Position | Origin | Destination | Repetition |
|---|---|---|---|
| 1604 | AATGAC | A—AC | 1552 |
| 686 | AAGTCATTT | ——— | 237 |
| 21989 | GTTTAT | GT—T | 69 |
| 515 | GTTATG | —— | 56 |
| 506 | CATGGTCATGTTATGGTT | CA—————T | 54 |
| 6329 | TCAAATTCG | ——— | 44 |
| 509 | GGTCATGTTATG | G———TG | 43 |
| 28089 | GGTTCTAAA | G——AA | 26 |
| 21980 | TTTTTGGGTGTTTAT | TT—————T | 24 |
| 671 | TACGGCGCCGATCTA | T—————TA | 23 |

insertion mutations were found to be rare, accounting for less than 0.28% of all nucleotide mutations. However, while rare, such mutations can profoundly affect the sequence of viral proteins encoded by the SARS-CoV-2 genome.

Substitution mutations may be silent, which corresponds to the case where the codon changes but it does not alter the encoded amino acid. This is possible since most amino acids can be encoded by more than one distinct codon. Thus, if codon mutations result in a degenerate codon for a specific amino acid, changes will not be observed at the level of amino acids, and consequently, at the protein level. In Fig. 6, we compare the number of silent mutations to the total number of mutations for each codon in the SARS-CoV-2 genome, obtained from the GISAID data set. We see that, on average, silent mutations make up a small number of total mutations for most of the codons. However, some codons have a tendency to undergo mostly silent mutations. For example, the codon TTC was observed to undergo a total of 63,555 mutations, and more than 98.7% of them are silent.

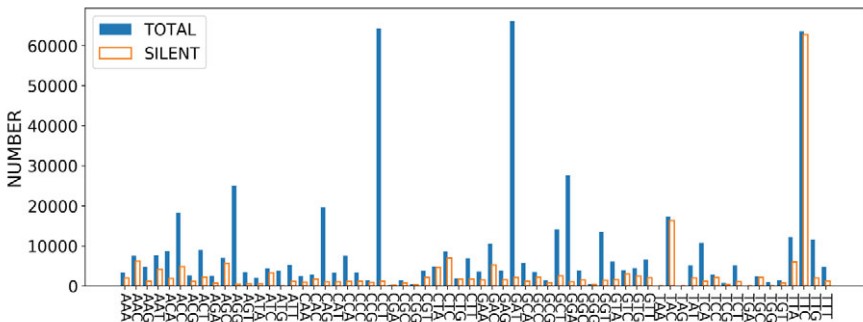

**Figure 6.** Total number of mutations (blue solid bar) and the number of silent mutations (orange bar) for distinct codon types, based on GISAID data set. The codons are arranged in alphabetical order along the horizontal axis.

### Text-mining methods identify conserved genome segments and novel therapeutic targets for SARS-CoV-2

Using the word ranking technique described in section 'Word ranking technique', we sought to determine the importance of different codons in the SARS-CoV-2 genome and use this to quantify the importance of each gene. To this end, we began by investigating the positions of each codon in the reference genome, which are depicted in Fig. 7 for the codons CGG and TAA. We see that while these two codons have approximately the same frequency of occurrence (11 and 10 times, respectively), their distribution along the genome is markedly distinct, with the codon CGG uniformly appearing in only three distinct genes. In contrast, the codon TAA appears towards the end of most coding regions in the genome and forms a cluster towards the end. These differences in the distribution of positions complies with the importance of the stop codon, which plays a key role in protein synthesis by terminating the decoding process.

To further quantify how the distribution of codon positions is related to the importance of the codon in the genome, we next calculated the eccentricities for all codons in the SARS-CoV-2 reference genome, which are depicted in Fig. 8. We see that the three codons GGG, CCC and GTC have the highest value of normalised eccentricity, $1.87 \times 10^{-2}, 1.82 \times 10^{-2}$ and $1.77 \times 10^{-2}$, respectively. In Fig. 8, we have ordered the codons based on their frequency of occurrence in the genome, which is also shown in

the plot for comparison. We also present the importance of each codon in the plot, calculated using Eq. (2) for the reference genome, which depends on both the eccentricity and the frequency of occurrence in the genome.

As we saw in Fig. 4, almost all occurrences of codons in different positions of the genomic sequence experience at least one mutation in the GISAID data set. Given the non-uniform distribution of mutation numbers across nucleotide positions, it is useful to classify the codons into low repetition and high repetition mutation groups. This is performed by segregating the codon-position pairs with a mutation repeat less than a threshold value into the low repetition group and the remaining pairs into the high repetition group. Here we take the threshold value to be eight, which leads to two groups with nearly equal number of codons.

Then, to quantify the degree of mutations in all codon-position pairs along the SARS-CoV-2 genome, we define the relative density of a codon $w$, denoted by $r(w)$, as follows:

$$r(w) = \frac{f_{\text{low}}(w)}{f(w)}. \tag{3}$$

In this last expression, $f_{\text{low}}(w)$ is the number of positions in which codon $w$ has a low number of mutations (defined to be less than eight in this work), and $f(w)$ is the frequency of occurrence of codon $w$ in the genome, as defined previously. Since a codon-position pair is either mutated a low number of times or a high

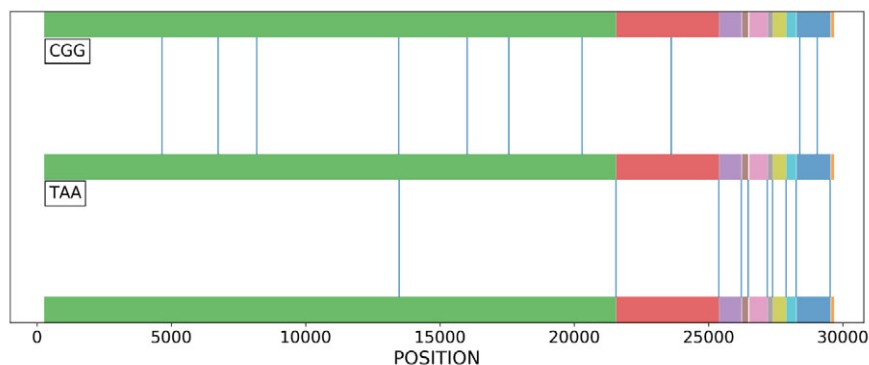

**Figure 7.** The positions of two codons, CGG and TAA, in the SARS-CoV-2 reference genome. The vertical blue lines are the position of the codons and the coloured rectangles are gene regions in the sequence. The two codons have almost the same frequency of occurrence, 11 and 10, respectively; however, their position along the genome is markedly different. TAA is the stop codon and plays an important role in protein-making instructions.

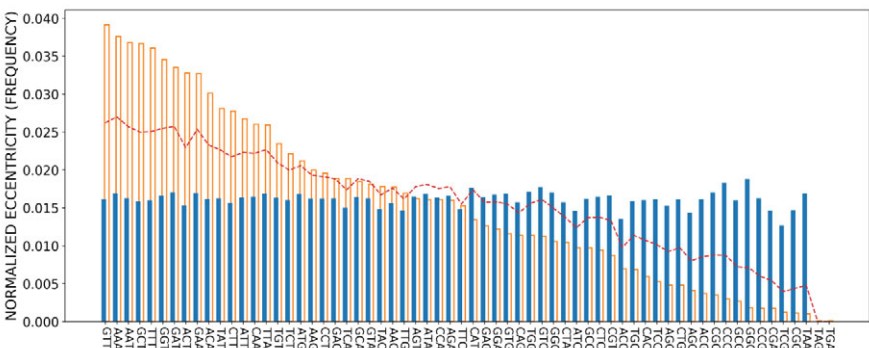

**Figure 8.** The eccentricity (solid blue bar) and frequency of occurrence (orange bar) for all the codons in the SARS-CoV-2 reference genome. The codons are arranged according to their frequency of occurrence, from most to least frequent. The red dashed line shows the codon importance, which takes into account both the normalised eccentricity and normalised frequency according to Eq. (2).

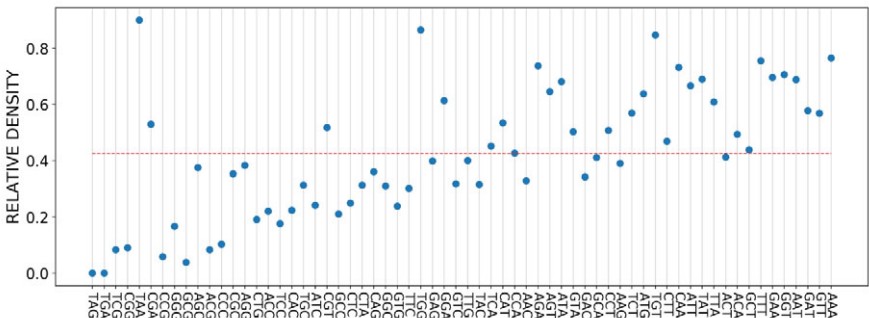

**Figure 9.** Relative density of codons in the SARS-CoV-2 genome, arranged in order of increasing codon importance, for the low mutation repetition group. Calculations were performed for the GISAID data set.

number of times, the relative density varies between zero and one for each codon $w$, where a value of one indicates that all codon-position pairs for codon $w$ experience a low number of mutations.

We plot the relative density of codons in Fig. 9, obtained by applying Eq. (3) to the GISAID data set. In the figure, the codons are arranged from least to most important, where the codon importance was calculated by applying Eq. (2) to the SARS-CoV-2 reference genome. From the figure, it is clear that there is a strong positive correlation between codon importance and relative density. To quantify the trend, we calculated the Pearson correlation between the codon importance and its relative density in the low repetition group, and found it to be greater than 0.68. Interestingly, these results imply that mutation information can be inferred directly from the codon importance values associated with the reference genome.

To validate the results, we can compare them with the results obtained from a random sequence with the same distribution of nucleotides and genomic structure. In a random sequence, the frequency of appearance of a three nucleotide segment is proportional to the product of the probability of its constituents, that is $f_{\mathrm{ran}}(w = x_1 x_2 x_3) \sim p(x_1)p(x_2)p(x_3)$. We can eliminate the effect of bias in usage of nucleotides by replacing the frequency in Eq. (2) with the relative frequency of a codon, $f(w)/f_{\mathrm{ran}}(w)$. This reduces the Pearson correlation between importance and relative density in the reference sequence to nearly 0.5. If we repeat the same calculation with a random sequence, the Pearson correlation becomes very close to zero. This implies that the codon

arrangement in the reference sequence is not random and instead obeys a kind of order.

Building on these results, we next ranked the SARS-CoV-2 genes according to the average of the relative density and importance of their constituent codons. The results are plotted in Fig. 10 and show that the two ranking schemes are strongly correlated with each other. To quantify the correlation, we calculated the Pearson correlation coefficient for the two ranked lists, which gave a value of greater than 0.91. These results confirm that the importance of viral genes, calculated from a reference genome using the text-mining methods developed in this work, can be used to infer which genes have a higher probability for mutation and those which are likely to be conserved. Furthermore, strong correlations are also observed between these two measures for the coding regions of the genome, as depicted in Supplementary Information 2, Fig. S3. The value of the Pearson correlation coefficient in this case is 0.90.

Given that text-mining methods were successful at pinpointing important mutation information for the viral genes, we next sought to determine if empirical laws that arise in linguistics are exhibited in viral genome sequences. To this end, we investigated how the frequency of codon-position mutations are related to their rank. As mentioned above, distinct codons can appear in different locations in the genome sequence, and in each location the codon experiences a different number of mutations. Therefore for each codon, we associate a set that is comprised of the number of its mutations in all its locations in the genome. We then ranked the values of each set in descending order for each codon. Doing so, we find that Zipf's law

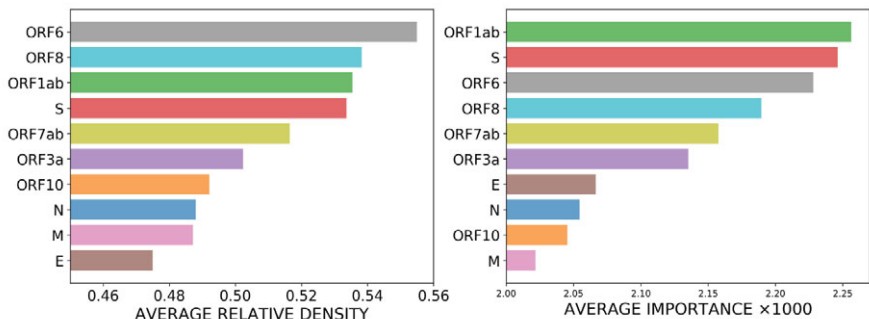

**Figure 10.** Average relative density and average importance of SARS-CoV-2 viral genes. The former is obtained from mutation data in GISAID data set and the latter is calculated based on the SARS-CoV-2 reference genome.

(Newman, 2005) holds between the number of mutations and the rank for all codons, implying there is a power law relation between these two quantities, as illustrated in Fig. 11. These results enable the quantification of the relative likelihood of different mutation events for each codon. Specifically, our findings indicate that the mutation event with rank $n$ appears $1/n^{\zeta}$ times as often as the most frequent mutation event for each codon, where $\zeta$ is called the Zipf exponent and is usually close to one.

The above result can be interpreted in the context of population genetics. Here, we consider mutations for a certain codon and assume there is only one mutation in each sequence. Therefore, the mutated sequence plays the role of an allele for the reference sequence. It has been previously shown that the frequency distribution of alleles obeys a power law relationship like the Zipf's law (Rothman and Templeton, 1980).

Finally, given that each codon position is associated with a different number of mutations, we sought to define a statistical quantity, which we refer to as the mutation index, that characterises the overall set of mutations for each codon type. Since the set of mutation values for each codon follows Zipf's law, we chose to take the median mutation number as the mutation index. In Fig. 12, we plot the mutation index of each codon in relation to the codon importance. Importantly, we can see from the figure that increasing importance values correspond to a lower mutation index. This implies that there is a strong negative correlation between these two quantities. To quantify the correlation, we calculated the Pearson correlation coefficient between the codon mutation index and importance values and found it to be greater than 0.70 in magnitude.

The mutation index is defined for any specific part of genome by averaging the mutation indices of the constituents of that segment. Using this approach, we next calculated the mutation index, based on the mutation data from the GISAID data set, for each of the SARS-CoV-2 genes in the reference genome. Similarly, we calculated the importance of each gene by taking the average of the importance values of their constituent codons in the reference genome. To quantify the strong negative correlation between these two measures, we calculated the Pearson correlation coefficient, which had a value of $-0.92$. To illustrate the negative correlation, we plot the average mutation index and the average importance of the SARS-CoV-2 viral genes in Fig. 13. In the figure, the genes are ranked in order of increasing mutation index and decreasing importance.

Notably, the average mutation index and the average importance of the coding regions of the SARS-CoV-2 viral genes are also highly correlated. To quantify the degree of correlation, we calculated the Pearson correlation coefficient between these two measures for all of the coding regions in the SARS-CoV-2 genome, and found it to be $-0.91$, indicating a strong negative correlation with a similar magnitude as the correlation for the viral genes. Thus, we can rank the genes either from most to least important, or by increasing mutation index, and we find that the difference between the two ranking schemes is negligible for most coding regions (see Supplementary Information 2, Fig. S4).

These results indicate that the importance of codons is strongly negatively correlated with their propensity for mutation. Thus, by analysing the importance of any segment of a reference genome, we can immediately infer the mutability of this segment. Specifically, in the case of novel viruses and pathogens, we can compute an index of mutability by calculating the importance of different segments from the first identified genome, without having to wait to gather genomes from other infected individuals to analyse the mutations. This is a significant finding because by requiring information about only one infection, the time it takes to develop targeted therapeutics for a novel pathogen is drastically reduced. Our findings can immediately be used to develop stable drugs that are based on short interfering RNA (siRNA) by targeting important genome segments of the pathogen that have low propensity for mutation. To this end, we present the top 10 important segments which consist of six and seven codons, respectively, in Tables 4 and 5. Inspection of Tables 4 and 5 confirms that non-structural protein 3 (nsp3) and spike protein (S) are the best targets for antiviral drugs (Lei *et al.,* 2018; Angeletti *et al.,* 2020; Frick *et al.,* 2020; Ong *et al.,* 2020; Pang *et al.,* 2020). It should be noted that in Tables 4 and 5, we have scaled the importance values of the codons between 0 and 1 using the formula:

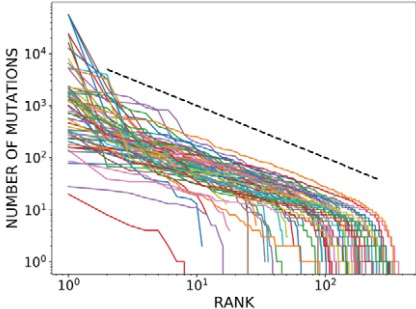

**Figure 11.** Number of mutations *versus* rank for all codon positions in the genome, for each codon type, obtained from the GISAID data set. Different codons are distinguished by different colours. We observe that Zipf's law holds for all codons. The black dashed line corresponds to a power law function with the form $y \propto x^{-1}$.

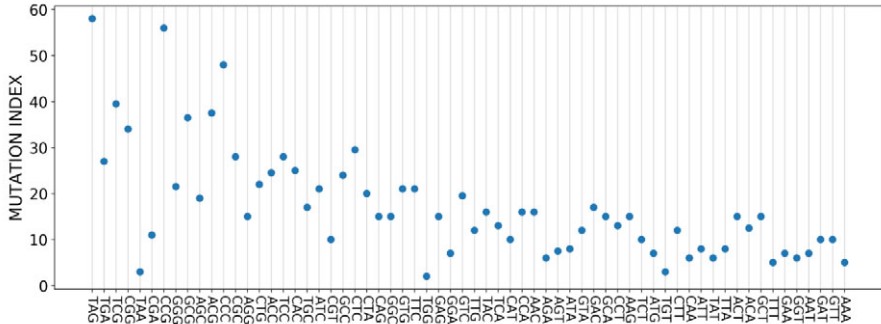

**Figure 12.** Mutation index of each codon, with codons arranged from least to most important. The plot depicts a strong negative correlation between the mutation index and the codon importance.

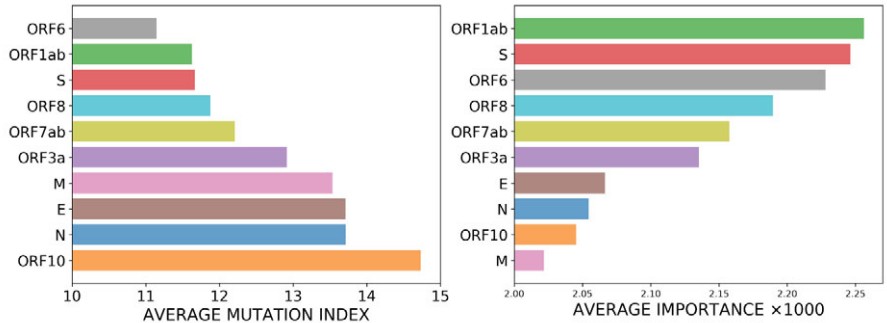

**Figure 13.** Average mutation index and average importance of SARS-CoV-2 viral genes. The former is obtained from mutation data in GISAID data set and the latter is calculated based on the SARS-CoV-2 reference genome. To illustrate negative correlation we rank the genes from the lowest to the highest average mutation index (left panel) in contrast to the ranking order for the importance values.

**Table 4.** Top 10 important segments with six codons in the SARS-CoV-2 genome sequence

| Segment | Position | Importance | Peptides | Coding region |
|---|---|---|---|---|
| AAAGTTGATGGTGTTGAT | 19720 | 0.9653 | KVDGVD | endoRNAse;ORF1ab |
| AAAAATGTTACAAAAGAA | 21067 | 0.9541 | KNVTKE | 2′-o-MT;ORF1ab |
| AATTTTAAAGTTACAAAA | 1787 | 0.9522 | NFKVTK | ORF1a;nsp2;ORF1ab |
| ACAAAAGTTGATGGTGTT | 19717 | 0.9502 | TKVDGV | endoRNAse;ORF1ab |
| AAAGATTTTGGTGGTTTT | 23945 | 0.9493 | KDFGGF | S |
| GAAACTAAAGATGTTGTT | 8204 | 0.9473 | ETKDVV | ORF1a;nsp3;ORF1ab |
| ACTAAAGATGTTGTTGAA | 8207 | 0.9473 | TKDVVE | ORF1a;nsp3;ORF1ab |
| GGTGTTGAAGGTTTTAAT | 23006 | 0.9462 | GVEGFN | S |
| AATGGTGTTGAAGGTTTT | 23003 | 0.9462 | NGVEGF | S |
| AATGTTACAAAAGAAAAT | 21070 | 0.9462 | NVTKEN | 2′-o-MT;ORF1ab |

$$i_{\text{scaled}}(w) = \frac{i(w) - i_{\min}}{i_{\max} - i_{\min}}, \qquad (4)$$

where $i_{\min}$ ( $i_{\max}$) is the minimum (maximum) importance value for the codons. Following the convention above, the importance of a segment is the average of the importance of its constituent codons.

## Discussion

In this work, we developed a simplistic but powerful text-mining method to identify the most important segments in a genome sequence. As shown in the work, the length of the segment is arbitrary, thus the method can be used to identify important

**Table 5.** Top 10 important segments with seven codons in the SARS-CoV-2 genome sequence

| Segment | Position | Importance | Peptides | Coding region |
|---|---|---|---|---|
| AAAAATGTTACAAAAGAAAAT | 21067 | 0.9538 | KNVTKEN | 2′-o-MT;ORF1ab |
| GGTAATTTTAAAGTTACAAAA | 1784 | 0.9510 | GNFKVTK | ORF1a;nsp2;ORF1ab |
| ACAAAAGTTGATGGTGTTGAT | 19717 | 0.9507 | TKVDGVD | endoRNAse;ORF1ab |
| AAAGATTTTGGTGGTTTTAAT | 23945 | 0.9498 | KDFGGFN | S |
| AATGGTGTTGAAGGTTTTAAT | 23003 | 0.9471 | NGVEGFN | S |
| GAAACTAAAGATGTTGTTGAA | 8204 | 0.9462 | ETKDVVE | ORF1a;nsp3;ORF1ab |
| GGTGGTAAAATTGTTAATAAT | 8549 | 0.9412 | GGKIVNN | ORF1a;nsp3;ORF1ab |
| GATTTTGGTGGTTTTAATTTT | 23948 | 0.9394 | DFGGFNF | S |
| ACTAAAAATGTTACAAAAGAA | 21064 | 0.9392 | TKNVTKE | 2′-o-MT;ORF1ab |
| ATTAAAGATTTTGGTGGTTTT | 23942 | 0.9317 | IKDFGGF | S |

segments across several length scales, including at the level of codons, genes, or gene coding regions. To illustrate the general applicability of the method, we implemented it to identify important segments in the SARS-CoV-2 reference genome (NCBI, 2020) directly from their frequencies of occurrence and spatial distribution along the sequence. Significantly, we showed that the segments that were identified as important strongly correlate with the conserved sequences identified through pairwise alignment of the reference genome with nearly 80,000 SARS-CoV-2 predecessor complete genome sequences.

In our empirical mutational analysis, we identified over 732,000 nucleotide mutations in the SARS-CoV-2 genome data set, and over 83% of these mutations were found to occur in coding regions of the virus, potentially leading to changes at the protein level. While characterising the nucleotide mutations, we found that the mutation frequency was negatively correlated with the frequency of occurrence of each nucleotide in the reference SARS-CoV-2 genome. Furthermore, we found that for each nucleotide, there was one distinct destination for substitution mutations, which were the most common type of mutations observed in the data set. Overall, nucleotide mutations tended to increase the AT content of the genome. Looking at the neighbours of mutated nucleotides, we further determined that some nucleotide sequences have a higher propensity for mutation than others. Notably, a nucleotide with a Guanine, Adenine, or Thymine, but not Cytosine, in its immediate vicinity was found to have a considerably higher probability of mutation, see Supplementary Information 1.

Analysing the spatial distribution of nucleotide mutations, we found that almost all nucleotide positions exhibited at least one mutation within the data set. However, certain positions along the genome exhibited a much higher rate of mutation, with the top five positions comprising nearly 40% of all the genomic mutations. Analysing the distribution of the number of codon mutations, we found that most codon positions in the genome have a small number of mutations, with the peak number being $\approx 16 - 32$, and considerably fewer sites have a significant ($\gtrsim 2^{11}$) number of codon mutations. Importantly, we found that for most codons, silent mutations, which do not lead to protein-level changes, make up a small number of total mutations. However, some codons experienced silent mutations almost exclusively. It should be noted that the mutation rate across the total number of nucleotides in the genome and the number of samples in the data set, overall, was fairly low compared with some other RNA viruses (Drake *et al.*,

1998). However, these results are crucial for understanding the diversity of the SARS-CoV-2 viral proteins, which is necessary for the development of effective vaccines and therapeutic strategies (Mercatelli and Giorgi, 2020).

We introduced two new measures in this work, the relative density and the mutation index, to characterise the mutations observed in different segments of the genome. We found that the relative density of a genome segment in the low mutation group was strongly correlated with its importance value, which indicated that important segments tended to have a low number of mutations. Significantly, this correlation was observed to hold at the codon and gene levels, and also for the coding regions of each gene. Importantly, we found that Zipf's law holds between the number of mutations of a codon position and its rank in the set of mutations for each codon. This enabled us to define the mutation index to characterise the mutability of different codons. We found that the mutation index of different genome segments was highly negatively correlated with the importance of these segments, with the strong correlation observed at the codon and gene levels, as well as for the coding regions of each gene. This strengthens the conclusion that important segments have a lower number of mutations. Notably, these findings are important for identifying potential candidates for stable siRNA-based targeted drugs that can inhibit the production of viral proteins. To illustrate this point further, we used the text-mining method developed in this work to identify the most important six and seven codon sequences from the reference SARS-CoV-2 genome, which are most likely to be stable against future genomic mutations and may therefore be candidates for siRNA-based antiviral drugs.

Crucially, the SARS-CoV-2 genes that were identified by the text-mining approach developed here confirm previous findings into the pathogenesis of the virus in humans (Mercatelli and Giorgi, 2020). Specifically, the genes encoding the structural proteins (S, E, M, N) and the gene ORF1ab, which encodes several non-structural proteins, were identified to be important by application of the method to the reference SARS-CoV-2 genome. Indeed, these proteins are thought to play a crucial role in the pathogenesis of the virus (Yoshimoto, 2020). In addition, genes that encode accessory proteins (ORF6, ORF8, ORF7ab, ORF3a and ORF10) were identified as highly important by our method. These proteins are thought to play a role in counteracting the host's innate immune system (Yoshimoto, 2020). The SARS-CoV-2 virus is known to induce an innate immune response, including the release of pro-inflammatory cytokines such

as TNF- *α*, IL-1 and IL-6 (Vabret *et al.,* 2020). This inflammatory response can lead to a cytokine storm, resulting in severe COVID-19 disease conditions and a high fatality rate. It is important to understand more about how variants of the accessory proteins are linked to innate immune signalling and severe disease outcome, and this will be investigated in a future work.

Severe COVID-19 disease conditions may also be related to the development of autoimmunity due to homology between the viral proteins and human proteins (Lyons-Weiler, 2020). This is a pivotal consideration when searching for antiviral targets and developing new vaccines. In recent work, it was determined that all of the SARS-CoV-2 proteins with immunogenic peptides have at least one match to human proteins; however non-human-like epitopes have also been identified (Lyons-Weiler, 2020; Sørensen *et al.,* 2020). Importantly, when our method was used to identify genomic segments with six codons as potential targets for siRNA-based therapeutics, one of the identified segments corresponds to a non-human-like epitope of the SARS-CoV-2 spike glycoprotein (Sørensen *et al.,* 2020). Thus, our findings may immediately be realisable for the development of an siRNA-based therapeutic that can target an epitope of a structural SARS-CoV-2 protein, without the risk of inducing an autoimmune response and severe disease outcome.

In closing, we note that the text-mining method developed here is generic and it enables the rapid identification of segments of a whole genome that are likely to remain conserved during future genomic mutation events. Importantly, these segments are identified from a reference (ancestral) genome. Thus, the method eliminates the need to wait for the collection and analysis of predecessor whole genome sequences. This not only reduces the cost, but is also crucial for the timely response to highly infectious novel pathogens that have the potential to cause widespread epidemics or global pandemics. Importantly, the approach can be applied to any pathogen, including for the identification of novel therapeutic strategies to help overcome antimicrobial resistance, which is considered one of the biggest threats to global health. This direction of research will be considered in a future work.

**Financial support.** The authors acknowledge the financial support from the Canadian Institutes of Health Research (CIHR).

**Conflict of interest.** The authors declare no conflicts of interest.

**Author contributions.** All authors have equal contribution to the work.

**Supplementary Materials.** To view supplementary material for this article, please visit http://dx.doi.org/10.1017/qrd.2021.13.

**Open Peer Review.** To view the open peer review materials for this article, please visit http://dx.doi.org/10.1017/qrd.2021.13.

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
