## [Reviewer Report]

*Comments to Author*: Darooneh et al. report on application of text-mining techniques for genomic analysis of the SARS-CoV-2 virus. The authors have analyzed two genomic datasets and identify genomic contexts and regions with differential mutability. The paper is well written and the results are interesting.

Major comments:

- What percentage of sequences in the NCBI and GISAID databases are common between them? (They are not independent databases.) Is it possible that high corroboration between the datasets is because of a large overlap? (If overlap is large, the analysis should be performed on either the union or intersection of datasets.)

- Throughout the paper, statements need quantification and numbers should be provided in the text. For instance, on page 9, in "a nucleotide has a considerable probability..." the probability must be stated. Qualitative words (such as several, rare, small, etc.) should be quantified throughout.

- Related to the previous comment, the text should describe the results. For instance, on page 10, this sentence is pointing to where the results are, but does not describe them: "several of the nucleotide positions with high probability for mutation in Table I are apparent in Fig. 4 as individual or small clusters of peaks in the number of mutations." Moreover, it is not clear where in Figure 4 this information can be found.

- More than 97% of the viral genome is coding its genes; therefore, it’s expected that 83% of the mutations are coding. Throughout the paper, normalization for lengths of the genome or individual genes need to be considered and explicitly stated so the significance of the findings can be assessed vs. random.

- There are known relationships between Zipf ’s law and measures in population genetics such as the infinite-allele model. The findings need to be discussed within what is known about SARS-CoV-2’s evolution and fixed mutations in the genomes analyzed.

- More insight in behavior of "importance i(w)" would help with clarity and justification of logarithms and square roots. How does this approach compare with normalization and weighting approaches to reduce large differences?

- TTC/TTT code for phenylalanine; more than 98% of mutations in this codon have to be C>T on the last base so they don’t change the amino acid. Were they corrected for codon usage in the viral genome and were the 63,555 mutations in TTC unique or were they the same mutation found in many viral genomes suggesting fixation? How were the numbers in Figure 6 calculated exactly?

- How are results in Figure 8 inform on codon usage in SARS-CoV-2’s genome?

- Loss of GC and increase in AT content is known for many human viruses as they evolve. Many papers exist on the topic, including https://journals.plos.org/plospathogens/article?id=10.1371/journal.ppat.1000079

Minor comments:

- Are codons defined based on how they code the proteins or a word is three consecutive nucleotide? How is a word defined for non-coding regions?

- The presented results point to a mutagenesis process, possibly due to antigenic drift. How can these findings be translated to other novel viruses and pathogens as suggested in Discussion when different mutagenesis processes may be involved?

- Are highly mutated regions known to be subjected to antigenic pressures?

- Can the rankings presented for importance of the genes be discussed in what is now known of their biology?- In multiple figures, the x axes are codons that are arranged based on various criteria. These criteria and ranking should be stated in the figure itself.

- This sentence is not clear on page 5: "Like the keywords in text, we assume that the significant codons form clusters."

- The phrase "experience mutation" should be replaced with "are mutated" throughout the paper.

---

## [Reviewer Report]

*Comments to Author*: Reviewer #1: Darooneh et al. report on application of text-mining techniques for genomic analysis of the SARS-CoV-2 virus. The authors have analyzed two genomic datasets and identify genomic contexts and regions with differential mutability. The paper is well written and the results are interesting.

Major comments:

- What percentage of sequences in the NCBI and GISAID databases are common between them? (They are not independent databases.) Is it possible that high corroboration between the datasets is because of a large overlap? (If overlap is large, the analysis should be performed on either the union or intersection of datasets.)

- Throughout the paper, statements need quantification and numbers should be provided in the text. For instance, on page 9, in "a nucleotide has a considerable probability..." the probability must be stated. Qualitative words (such as several, rare, small, etc.) should be quantified throughout.

- Related to the previous comment, the text should describe the results. For instance, on page 10, this sentence is pointing to where the results are, but does not describe them: "several of the nucleotide positions with high probability for mutation in Table I are apparent in Fig. 4 as individual or small clusters of peaks in the number of mutations." Moreover, it is not clear where in Figure 4 this information can be found.

- More than 97% of the viral genome is coding its genes; therefore, it’s expected that 83% of the mutations are coding. Throughout the paper, normalization for lengths of the genome or individual genes need to be considered and explicitly stated so the significance of the findings can be assessed vs. random.

- There are known relationships between Zipf ’s law and measures in population genetics such as the infinite-allele model. The findings need to be discussed within what is known about SARS-CoV-2’s evolution and fixed mutations in the genomes analyzed.

- More insight in behavior of "importance i(w)" would help with clarity and justification of logarithms and square roots. How does this approach compare with normalization and weighting approaches to reduce large differences?

- TTC/TTT code for phenylalanine; more than 98% of mutations in this codon have to be C>T on the last base so they don’t change the amino acid. Were they corrected for codon usage in the viral genome and were the 63,555 mutations in TTC unique or were they the same mutation found in many viral genomes suggesting fixation? How were the numbers in Figure 6 calculated exactly?

- How are results in Figure 8 inform on codon usage in SARS-CoV-2’s genome?

- Loss of GC and increase in AT content is known for many human viruses as they evolve. Many papers exist on the topic, including https://journals.plos.org/plospathogens/article?id=10.1371/journal.ppat.1000079

Minor comments:

- Are codons defined based on how they code the proteins or a word is three consecutive nucleotide? How is a word defined for non-coding regions?

- The presented results point to a mutagenesis process, possibly due to antigenic drift. How can these findings be translated to other novel viruses and pathogens as suggested in Discussion when different mutagenesis processes may be involved?

- Are highly mutated regions known to be subjected to antigenic pressures?

- Can the rankings presented for importance of the genes be discussed in what is now known of their biology?- In multiple figures, the x axes are codons that are arranged based on various criteria. These criteria and ranking should be stated in the figure itself.

- This sentence is not clear on page 5: "Like the keywords in text, we assume that the significant codons form clusters."

- The phrase "experience mutation" should be replaced with "are mutated" throughout the paper.

---

## [Reviewer Report]

*Comments to Author*: The authors developed an approach to calculate the importance of a segment in a sequence based on a text-mining approach. The importance of a segment is a combination of the frequency and eccentricity. Then the authors derived the relative density and mutational index as quantities of mutational behavior. As an application, the authors have performed substantial work to calculate descriptives and investigate the mutational properties of the SARS-CoV-2 virus, as well as use their measures to interpret the findings.

Major comments:

1. I have difficulties with the eccentricity, as defined in [Disp-formula eqn1]. The authors describe it as ‘clustering of a codon’ (page 7, line 16), and more generally ‘clustering of a word’ (page 6, line 27).

a. The authors need to clearly define what ’clustering of a word/codon’ precisely means, do they mean the clustering of several codons? Or the clustering of several occurrences of a single codon? E.g. are two clustered codons on average located closer to each other than two other arbitrary other codons? Or are the occurrences of a clustered codon on average closer to each other than for another codon?

b. [Disp-formula eqn1] should capture clustering as well as distance from the first quarter of a region. In the equation, the authors sum the squared position of a codon minus the position of the first quarter. The authors should explain why the sum of squared positions capture clustering. It seems that the Equation puts more emphasis on whether a segment is far from the first quarter, regardless of it being clustered. Namely, a codon that is clustered around the first quarter has almost no impact on the eccentricity, unless at least one occurrence is far away from the first quarter.

c. How do the authors deal with differences in region length? In a longer region, a codon can be located further away from the first quarter, yielding a higher eccentricity. This seems unfair for shorter regions.

2. I also have difficulties with the importance, defined in [Disp-formula eqn2]. I agree that the frequency and eccentricity should be normalized before they are combined in one measure. The authors chose to divide each e(w) and f(w) by the total eccentricity resp. frequency. This leads to a normalized quantity between 0 and 1. The authors also applied a log transformation and a square root to "reduce large differences" and account for a "larger scale of the frequencies". After dividing by the total frequency and eccentricity, these two issues seem not a problem anymore, as all values lie between 0 and 2. Moreover, taking the square root of numbers between 0 and 1 increases their scale. The authors need to justify taking the log and square root.

3. On page 4, last two lines, the authors describe the data used in this manuscript. How crucial is it to have many known sequences in order to apply their method? Can the mutability be reliably estimated when not so much is known about a virus? Also, with a very low or high mutability, do we need significantly more or less sequences to identify the important segments? Perhaps the authors can state something about this in the Discussion section.

Minor comments:

- Page 6, line 19. The term "finite probability" is odd, since a probability is by definition finite. Do the authors mean "equal probability" (or maybe "non-zero probability")?

- Page 7, line 17. It would help the reader if the authors would add a sentence explaining why clustering near region boundaries is more important.

- Page 19, line 4. Define normalized eccentricity. Is that the second part of [Disp-formula eqn2]?

- Page 19, line 7. The authors state that more frequent codons correspond with lower eccentricity. This cannot be derived from figure 8, as the eccentricity seems more or less equal across codons. The authors should weaken their statement.

---

## [Reviewer Report]

*Comments to Author*: Reviewer #2: The authors developed an approach to calculate the importance of a segment in a sequence based on a text-mining approach. The importance of a segment is a combination of the frequency and eccentricity. Then the authors derived the relative density and mutational index as quantities of mutational behavior. As an application, the authors have performed substantial work to calculate descriptives and investigate the mutational properties of the SARS-CoV-2 virus, as well as use their measures to interpret the findings.

Major comments:

1. I have difficulties with the eccentricity, as defined in [Disp-formula eqn1]. The authors describe it as ‘clustering of a codon’ (page 7, line 16), and more generally ‘clustering of a word’ (page 6, line 27).

a. The authors need to clearly define what ‘clustering of a word/codon’ precisely means, do they mean the clustering of several codons? Or the clustering of several occurrences of a single codon? E.g. are two clustered codons on average located closer to each other than two other arbitrary other codons? Or are the occurrences of a clustered codon on average closer to each other than for another codon?

b. [Disp-formula eqn1] should capture clustering as well as distance from the first quarter of a region. In the equation, the authors sum the squared position of a codon minus the position of the first quarter. The authors should explain why the sum of squared positions capture clustering. It seems that the Equation puts more emphasis on whether a segment is far from the first quarter, regardless of it being clustered. Namely, a codon that is clustered around the first quarter has almost no impact on the eccentricity, unless at least one occurrence is far away from the first quarter.

c. How do the authors deal with differences in region length? In a longer region, a codon can be located further away from the first quarter, yielding a higher eccentricity. This seems unfair for shorter regions.

2. I also have difficulties with the importance, defined in [Disp-formula eqn2]. I agree that the frequency and eccentricity should be normalized before they are combined in one measure. The authors chose to divide each e(w) and f(w) by the total eccentricity resp. frequency. This leads to a normalized quantity between 0 and 1. The authors also applied a log transformation and a square root to "reduce large differences" and account for a "larger scale of the frequencies". After dividing by the total frequency and eccentricity, these two issues seem not a problem anymore, as all values lie between 0 and 2. Moreover, taking the square root of numbers between 0 and 1 increases their scale. The authors need to justify taking the log and square root.

3. On page 4, last two lines, the authors describe the data used in this manuscript. How crucial is it to have many known sequences in order to apply their method? Can the mutability be reliably estimated when not so much is known about a virus? Also, with a very low or high mutability, do we need significantly more or less sequences to identify the important segments? Perhaps the authors can state something about this in the Discussion section.

Minor comments:

- Page 6, line 19. The term "finite probability" is odd, since a probability is by definition finite. Do the authors mean "equal probability" (or maybe "non-zero probability")?

- Page 7, line 17. It would help the reader if the authors would add a sentence explaining why clustering near region boundaries is more important.

- Page 19, line 4. Define normalized eccentricity. Is that the second part of [Disp-formula eqn2]?

- Page 19, line 7. The authors state that more frequent codons correspond with lower eccentricity. This cannot be derived from figure 8, as the eccentricity seems more or less equal across codons. The authors should weaken their statement.